# A NEWBORN EMBODIED TURING TEST FOR COMPARING OBJECT SEGMENTATION ACROSS ANIMALS AND MACHINES

Manjulata Garimella[1], Denizhan Pak[1,3], Justin N. Wood[1,2,3,4], and Samantha M. W. Wood[1,3,4]

[1]Department of Informatics, School of Informatics, Computing, & Engineering
[2]Department of Psychological & Brain Sciences
[3]Cognitive Science Program
[4]Program in Neuroscience

## ABSTRACT

Newborn brains rapidly learn to solve challenging object perception tasks, including segmenting objects from backgrounds and recognizing objects across new viewing situations. Conversely, modern machine learning (ML) algorithms are "data hungry," requiring more training data than brains to reach similar performance levels. How do we close this learning gap between brains and machines? Here, we introduce a new benchmark—a Newborn Embodied Turing Test (NETT) for object segmentation—in which newborn animals and machines are raised in the same environments and tested with the same tasks, permitting direct comparison of their learning. First, newborn chicks were raised in controlled environments containing a single object rotating on a single background, then their recognition performance was tested across new backgrounds and viewpoints. Second, we performed "digital twin" experiments in which artificial agents were reared and tested in virtual environments that mimicked the rearing and testing conditions of the chicks. We inserted a variety of ML "brains" into the artificial agents and measured whether those algorithms learned common object recognition behavior as chicks. All newborn chicks solved this one-shot object segmentation task, successfully learning *background-invariant* object representations that generalized across new backgrounds and viewpoints. In contrast, none of the artificial agents solved the task, instead learning *background-dependent* representations that failed to generalize across new backgrounds and viewpoints. This digital twin design exposes core limitations in current ML algorithms in developing brain-like object perception. Our NETT is publicly available for comparing ML algorithms with newborn chicks. We argue that NETT benchmarks can help researchers build embodied AI systems that learn as efficiently and robustly as newborn brains.

## 1 INTRODUCTION

A longstanding goal in artificial intelligence (AI) is to build machines that learn like brains. Early AI research laid the foundation for biologically inspired, neurally mechanistic machine learning (ML) models, and recent progress in deep learning shows that AI algorithms can be used to 'reverse engineer' sensory processing in animals—including vision (Yamins & DiCarlo, 2016), audition (Kell et al., 2018), and olfaction (Wang et al., 2021)—while also informing higher-level abilities—including visually guided action (Michaels et al., 2020), language (Schrimpf et al., 2021), navigation (Whittington et al., 2022), and memory (Nayebi et al., 2021).

But how will we know when we have succeeded in building machines that learn like brains? We argue that an evaluation platform must meet two requirements: (1) animals and machines must be raised in the same environments; (2) animals and machines must be tested with the same tasks. The first requirement stems from the observation that the intelligence of a system depends both on its learning algorithms and the training data (experiences) it receives. Any observed differences in behavior across animals and machines could be due to differences in learning algorithms, training

data, or some combination of the two factors. Consequently, evaluating whether machines have similar learning algorithms as animals requires giving machines the same training data as animals. The second requirement follows from the observation that evaluations of intelligence and learning are task dependent. Accordingly, biological and artificial systems must be evaluated with the same tasks to ensure that any observed differences are not due to differences in the tasks themselves.

While these two requirements may seem straightforward, building experimental platforms that meet both requirements has not yet been possible. Controlling the training data across animals and machines requires performing parallel controlled-rearing experiments on animals and machines. However, most newborn animals cannot be raised in controlled environments from birth, preventing researchers from controlling the training data presented to animals. Accurate comparison between animals and machines also requires measurements with a high signal-to-noise ratio, where a subject's behavior in response to a particular stimulus (e.g., an object) can be reliably estimated (Wood & Wood, 2019). However, most studies with newborn subjects have produced noisy data (i.e., high measurement error), so researchers lacked the high precision data needed to make accurate comparisons across newborn animals and machines. Finally, the field lacked an experimental platform for raising machines in the same environments as newborn animals, preventing researchers from matching the training data across animals and machines.

Here, we present an approach—matched-experience Newborn Embodied Turing Tests (NETTs)—that overcomes these barriers, allowing newborn animals and machines to be raised in the same environments and tested with the same tasks (Fig.1). NETTs are inspired by Alan Turing's goal of creating tests for determining whether AI systems are capable of human-like intelligence. In the original Turing test, if a person cannot determine whether they are speaking to an AI system or a human, the AI passes the test (Turing, 1950). Turing's original test focused on linguistic competence, and large language models can now pass this test (Brown et al., 2020). Similarly, NETTs assess whether AI systems are capable of mimicking newborn intelligence by performing parallel controlled-rearing studies on animals and machines. NETTs use a "digital twin" design in which artificial animals are raised in virtual environments that mimic the rearing and testing conditions of newborn animals. An artificial animal passes the NETT if its behavioral performance is indistinguishable from that of its biological counterpart. Rather than focusing on linguistic competence, NETTs focus on building embodied 'pixels-to-actions' machines that learn the same core competencies as animals when provided with the same experiences as animals. NETTs thus focus attention on the early emerging sensorimotor capacities that serve as the foundation for more advanced abilities learned later in life.

## 1.1 CHICKS AS AN ANIMAL MODEL FOR NETTs

NETTs can be created for any animal, provided that it is possible to control the animal's environment from birth and create digital twins (virtual replicas) of their environments for rearing and testing artificial animals (Lee et al., 2021a;b; McGraw et al., 2023; Pak et al., 2023). However, most commonly used animal models in psychology (e.g., primates, rodents, pigeons), cannot be raised in strictly controlled environments from birth. One species that can independently care for itself at birth is chickens (*Gallus gallus*). Therefore, it is possible to raise newborn chicks in controlled environments (e.g., environments void of all objects and caregivers; Wood & Wood 2015) and raise/test machines in virtual replicas of those environments (Pak et al., 2023).

Another advantage of using chicks as a NETT animal model is that researchers can study visual learning in chicks with high precision using automated controlled-rearing methods. Automated controlled-rearing experiments produce data with a high signal-to-noise ratio (Wood & Wood, 2019), which allows researchers to build NETTs with high noise ceilings for comparing learning across animals and machines.

Finally, studies of chicks can inform human development because avian and mammalian brains share many characteristics. On the circuit level, avian and mammalian brains contain homologous cortical circuits for processing sensory input (Karten, 2013). While these circuits are organized differently in birds and mammals (nuclear vs. layered organization, respectively), the circuits share similarities in terms of cell morphology, the connectivity pattern of the input and output neurons, gene expression, and function (Calabrese & Woolley, 2015; Dugas-Ford et al., 2012; Jarvis et al., 2005; Wang et al., 2010). On the architectural level, avian and mammalian brains share the same large-scale organizational principles: both are modular, small-world networks with a connective core of hub nodes

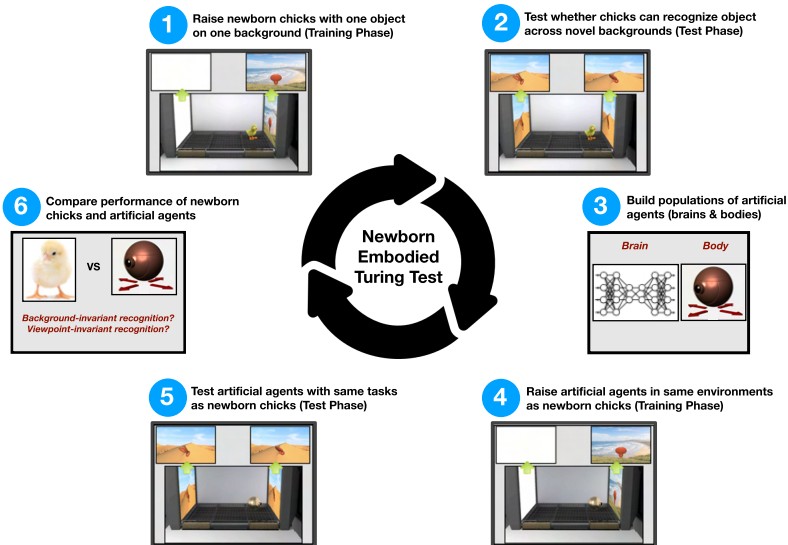

Figure 1: Benchmark for comparing the learning abilities of newborn animals and machines. Newborn chicks and artificial agents are raised in the same environments and tested with the same tasks. This Newborn Embodied Turing Test (NETT) evaluates whether animals and machines develop the same object segmentation abilities when provided with the same visual experiences. NETTs provide a closed-loop system for studying the origins of intelligence: researchers build ML models that learn like newborn animals, then use those models to generate predictions for new experiments with animals to validate and refine the models.

that includes visual, auditory, limbic, prefrontal, premotor, and hippocampal structures (Shanahan et al., 2013). Given the similarities between avian and mammalian brains, controlled-rearing studies of newborn chicks can inform our understanding of both avian and mammalian intelligence (see Section A.1 for detailed discussion).

## 1.2   ONE-SHOT OBJECT SEGMENTATION

In this paper, we created a NETT benchmark for testing whether AI algorithms can mimic the object segmentation and recognition abilities of newborn chicks. We focused on object segmentation because (1) it is required for real-world vision, with a long history of research in computer vision (Long et al., 2015; Redmon et al., 2016; He et al., 2017; Xie et al., 2021), (2) it is a complex computational feat, requiring the integration of many regions with different hue and luminance values into unified object representations (Ostrovsky et al., 2009), and (3) it is one of the earliest high-level visual abilities that emerges in newborn animals (Wood & Wood, 2021). For example, controlled-rearing studies show that newborn chicks can segment the first object they see from natural backgrounds, even when the chicks have received sparse visual training data (e.g., visual experience of a single object rotating on a single background, Wood & Wood 2021).

Our NETT tested whether newborn chicks and artificial agents are capable of one-shot object segmentation (Fig. 2). The subjects were presented with a challenging task. During the Training Phase, their visual world contained a single object rotating on a single background. As a result, from a statistical perspective, the object and background had a 100% concurrence rate. During the Test Phase, we tested whether the subjects could recognize the Training Phase object when the object was presented on novel backgrounds and from novel viewpoints.

If subjects depend solely on statistical learning to segment and recognize objects, then they should incorrectly bind the object and background features together because the object and background always co-occurred during training. In this case, the subject's object recognition performance would be high when the object is presented on the familiar background and low when the object is presented on novel backgrounds. Conversely, if subjects are capable of one-shot object segmentation (i.e., they can parse objects from backgrounds despite the 100% concurrence rate during training), then their recognition performance should be high, regardless of whether the object is presented on familiar or novel backgrounds.

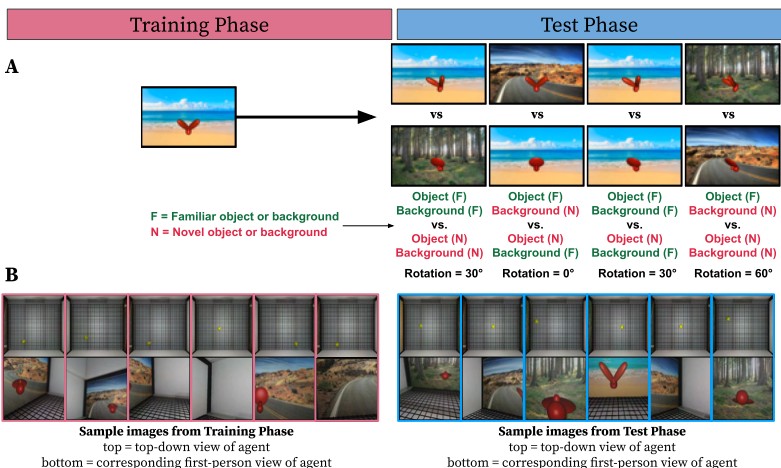

Figure 2: (A) One-shot object segmentation task. Newborn chicks and artificial agents are reared in environments containing a single object rotating on a single background, then tested on their ability to recognize that object across new backgrounds and viewpoints (changes in azimuth rotation). (B) Sample images from the Training and Test Phases from the artificial chick experiments.

## 2 METHODS

### 2.1 ANIMAL EXPERIMENTS

We focused on the object segmentation/recognition task and behavioral data from (Wood & Wood, 2021) (Exp. 1). All chick experiments and data are originally reported in Wood & Wood (2021). In the study, chicks (n = 35) were hatched in darkness, then raised singly in automated controlled-rearing chambers that measured each chick's behavior continuously (24/7) during the first 12 days of life. The chambers were equipped with two display walls (LCD monitors) for displaying object stimuli. The chambers did not contain any objects other than the virtual objects projected on the display walls. Thus, the chambers provided full control over all visual object experiences from the onset of vision.

During the Training Phase, chicks were reared in an environment containing a single 3D object rotating on a single background image. The object made a full 360° rotation every 15s along the horizontal axis. The chicks were reared with one of two possible objects, and the object was presented on one of three possible backgrounds. Due to filial imprinting, the chicks were naturally motivated to spend time with the object presented during the Training Phase. The chicks were raised in this sparse "one object, one background" environment for 5 days, allowing the critical period on filial imprinting to close.

During the Test Phase, the chambers measured the chicks' imprinting behavior and object recognition performance. The "imprinting trials" measured whether the chicks developed an imprinting response. During these trials, the familiar (imprinted) object was presented on one display wall and the other display wall was a blank white screen. If the chicks developed an imprinting response, then they should have spent more time by the familiar object than the blank display. The "object recognition trials" measured the chicks' ability to segment and recognize their familiar object. During these trials, the familiar object was presented on one display wall and a novel object was presented on the other display wall. Across object recognition trials, the familiar and novel objects were presented on all possible combinations of the three background (B) scenes (B1 vs. B1, B1 vs. B2, B1 vs. B3, B2 vs. B2, B2 vs. B3, and B3 vs. B3). For the evaluation, Wood & Wood (2021) grouped the test trials into four conditions (visualized in Fig. 2, bar colors in Fig. 3):

1. Familiar object on familiar background, novel object on novel background (*yellow bars*);

2. Familiar object on novel background, novel object on familiar background (*purple bars*);

3. Both objects on the familiar background (*blue bars*);

4. Both objects on a novel background (*green bars*).

The objects were also shown from three possible viewing angles: 0° change in azimuth rotation (same as Training Phase), 30° change in azimuth rotation, and 60° change in azimuth rotation. On each trial, the viewing angle of the imprinted and novel objects were matched. If the chicks developed object segmentation *and* the ability to recognize objects across novel backgrounds and viewpoints, then they should have spent more time by the display wall with the familiar object versus the novel object. Object recognition trials were scored as the proportion of time spent with the familiar (imprinted) object relative to the total time spent with the familiar and the novel object. Wood & Wood (2021) collected hundreds of object recognition trials from each chick by leveraging automated stimuli presentation and behavioral coding. As a result, the study produced data with a high signal-to-noise ratio (*red noise bands*, Fig. 3; Wood & Wood 2019).

## 2.2 MACHINE LEARNING EXPERIMENTS

Our goal was to raise and test biological and artificial agents under parallel conditions. This required (1) virtual environments for artificial agents that mimicked the environments faced by the chicks, and (2) artificial agents that could learn from raw sensory inputs, make decisions, and perform actions, akin to chicks.

**Virtual environments.** To simulate the visual environments of the chicks in Wood & Wood (2021), we created realistic digital twins (virtual replicas) of the controlled-rearing chambers, using a video game engine (Unity). The virtual chambers matched the proportions, textures, and colors of the real-world chambers (Section A.2). During the Training Phase and Test Phase, we reared and tested the artificial agents in the virtual chambers, presenting the same stimuli and tasks to the artificial agents that were presented to the chicks.

**Artificial agents.** To directly compare animals and machines, we created artificial agents that performed the same task as all newborn animals: learn from raw sensory data and perform actions in the environment. We created the artificial agents by embodying deep reinforcement learning (RL) algorithms in virtual chick bodies. Each body measured 3.5 units (height) by 1.2 units length (radius) and received visual input (64×64 pixel resolution images) through an invisible forward-facing camera attached to its head. The artificial agents could look up and down, move forward and backwards, rotate left and right, or remain stationary. The actions were represented as three continuous variables: (1) head movement up/down, (2) translation forward/backward, and (3) rotation around the vertical axis. To control the behavior of the artificial agents, we inserted a variety of artificial neural networks (ANNs) into the chicks (Section A.3, see Tables 1 and 2 in Supplementary Materials for ANN configurations). For each of the six rearing conditions in Wood & Wood (2021), we trained and tested five artificial agents per ML algorithm. All artificial agents in a group had the same network architecture, but each artificial agent's ANN started with a different random initialization of connection weights, and each artificial agent's connection weights were shaped based on its own particular experiences during the Training Phase.

**Imprinting Reward.** During real-world behavior, newborn chicks spontaneously reduce the distance between themselves and their imprinted object, making the imprinted object larger in their field of view. To encourage the artificial agents to imprint, the agents received a reward proportional to the size of the imprinted object in their field of view (details in Section A.4). For example, if the object extended through a quarter of the agent's field of view, then the reward for that step would be 0.25. The artificial agents were trained to optimize the sum of the imprinting reward using Proximal Policy Optimization (PPO; Schulman et al. 2017). As we show in SI Figures 5, 6, and 7, this imprinting reward produces similar imprinting behavior as biological chicks.

At the beginning of each training episode, the artificial agent was spawned at a random position and orientation within the chamber. Each training episode lasted 1,000 time steps. We trained the artificial agents for 1,000 episodes. After the Training Phase, the network weights were frozen for the Test Phase (i.e., the artificial agents did not receive any rewards during the Test Phase, and learning was discontinued). This freezing mimics the critical period of filial imprinting in biological chicks, in which chicks stop learning about their imprinted object after the first few days of life (Horn, 2004).

During the Test Phase (Section A.5), each artificial agent performed 1,080 object recognition trials (40 trials for each of the 27 background-viewpoint combinations presented to the chicks). At the beginning of each test episode, the artificial agent was spawned at the middle of the chamber, facing a blank wall. Each object recognition trial consisted of 1,000 time steps. At every time step, we recorded the position of the artificial agent in X,Y coordinates. As with the chicks, we measured

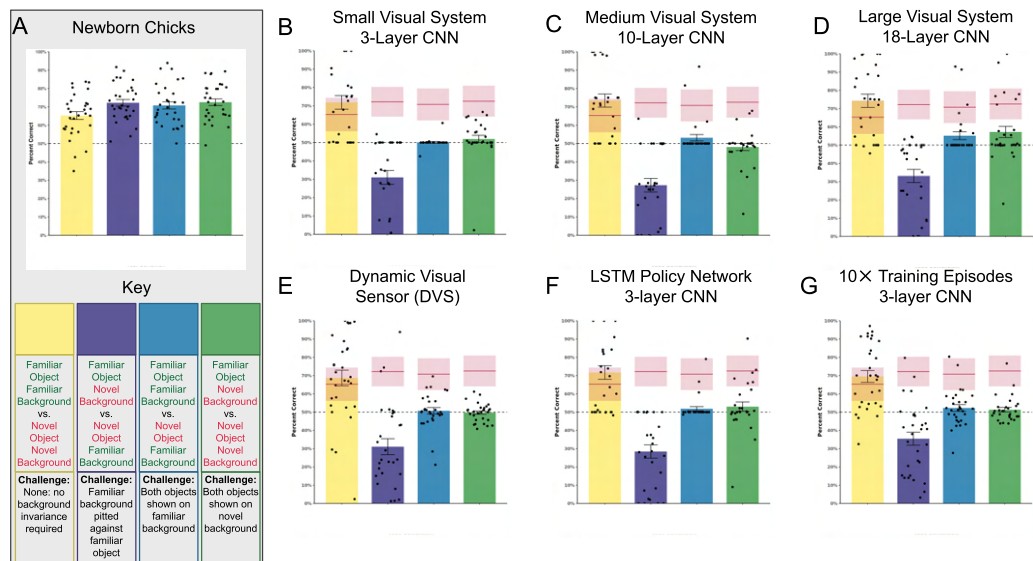

Figure 3: Performance of the newborn chicks and artificial agents on the object segmentation and recognition task. During the Test Phase, the familiar and novel objects were presented on familiar and novel backgrounds, in a fully crossed design (Key). (A) The chicks successfully recognized the object across all viewing conditions, regardless of whether the familiar and novel objects were presented on familiar or novel backgrounds. The chicks learned *background-invariant* object representations. (B-G) In contrast, the artificial agents learned *background-dependent* object representations, with the background features heavily influencing the recognition performance of the agents. To facilitate comparison between the chicks and artificial agents in Panels B-G, mean chick performance is shown as a red line with a "noise band" (red band) reflecting the average deviation of the chicks from the mean chick performance.

whether the artificial agent spent a greater proportion of time with the familiar (imprinted) object than the novel object.

## 3 RESULTS

### 3.1 BIOLOGICAL CHICK PERFORMANCE

On the imprinting trials, the chicks spent significantly more time by the imprinting stimulus than the blank screen ($M = 84\%$, $SD = 6\%$, $t(30) = 33.87$, $p < .0001$). On the individual level, all of the chicks successfully learned to imprint (range 68% to 93%, chance performance = 50%).

On the object recognition trials, the dependent variable was the proportion of time the chick spent by their familiar object versus the novel object. To compute performance, we used the following formula: *time by familiar object / (time by familiar object + time by novel object)*.

For the biological chicks (Fig. 3A), performance was well above chance level (50%) in all four background conditions (one-sample $t$-tests, all $Ps < 10^{-7}$, all Cohen's $ds > 1.2$). Performance was also well above chance level in all three viewpoint conditions (one-sample $t$-tests, all $Ps < 10^{-10}$, all Cohen's $ds > 1.7$). The chicks successfully recognized their familiar object, regardless of whether the object was presented on familiar or novel backgrounds or whether the object was presented from familiar or novel viewpoints. Thus, newborn chicks are capable of one-shot object segmentation.

### 3.2 ARTIFICIAL CHICK PERFORMANCE

All statistical analyses supporting claims in this section are in Supplementary Methods (Tables 3-5). We visualize the object representations learned by the agents using tSNE (Section A.6, SI Figure 2).

**Base Agents.** First, we examined the object recognition performance of the base agents, each of which was equipped with a simple 3-layer CNN visual encoder and a PPO policy learning algorithm. During the imprinting trials, all of the base agents learned to detect and approach the familiar object, indicating that all of the artificial agents learned to imprint.

During the test trials, however, the artificial agents showed an entirely different pattern of results from the chicks (Fig.3B). While the chicks recognized the familiar object across familiar and novel backgrounds, the artificial agents failed to recognize the familiar object across novel backgrounds. The artificial agents performed at chance level when both objects were shown on novel backgrounds and when both objects were shown on the familiar background. Unlike chicks, whose recognition performance is almost entirely invariant to background changes, the base agents learned background-dependent representations, incorrectly linking object and background features together.

**Medium and Large Visual Encoders.** The base agent had a small visual encoder (3-layer CNN), which may not have been sufficient to disentangle object features from background features during training. To test whether the background-dependent learning of the base agents was due to a small visual encoder, we equipped the artificial agents with medium (ResNet-10) and large (ResNet-18) visual encoders.

During the imprinting trials, all 30 of the agents with medium encoders, and 28 of the 30 agents with large encoders, learned to detect and approach the familiar object. However, despite having larger visual encoders than the base agents, the artificial agents developed the same form of recognition behavior (Fig.3C&D). The artificial agents all learned background-dependent object representations.

**Biologically Inspired Retinal Operations.** One way biological visual systems might disentangle object features from background features is through dynamic visual processing in the retina. Retinal ganglion cells compute temporal contrast (i.e., changes in luminance across time; Lichtsteiner et al. 2008. As a result, information passed from the retina to LGN is not in the form of static images, but rather visual events. When a still observer watches an object moving along a static background, only the contrast of the object will change. Visual information about the object will be enhanced, and the background will be suppressed.

To test whether retina-like processing can account for newborn chicks' ability to segment objects from backgrounds, we added retinal postprocessing to the images before sending the images to the ANN. Our retinal processing mimicked the operations of a Dynamic Visual Sensor (DVS) system (Posch et al., 2014; Gallego et al., 2020). For an image frame at time $t$: (1) convert pixels from RGB to grayscale/brightness, (2) apply a Gaussian blur, (3) compute the difference between the resulting matrix and the blurred grayscale matrix from $t$-1, (4) apply a threshold to all resulting matrix values.

During the imprinting trials, the artificial agents with DVS "eyes" learned to detect and approach the familiar object (26 out of 30 agents). However, despite having dynamic visual sensing, the artificial agents developed the same background-dependent recognition behavior as the prior agents (Fig.3E).

**Recurrent Learning Algorithms.** Another way biological visual systems might parse objects from backgrounds is through recurrent processing. Recurrent neural networks (RNNs) process dynamic, time-series data using feedback loops that allow prior information to persist. This form of short-term memory allows prior input to influence current inputs and outputs. To learn about the temporal dynamics of the visual environment, artificial agents might therefore need recurrent processing. To explore this possibility, we added recurrent processing to our base agents by converting their fully connected policy network into an LSTM (long short-term memory) network (Hochreiter & Schmidhuber, 1997). The LSTM network contained one LSTM layer with 256 hidden units.

During the imprinting trials, all 30 of the LSTM artificial agents learned to detect and approach the familiar object. However, despite having recurrent processing, the artificial agents developed the same form of background-dependent object recognition as the prior agents (Fig.3F).

**Longer Training.** All of the above agents were trained for 1,000 episodes. But some neural networks learn to "grok" a generalizable pattern from the data well after the point of overfitting (Power et al., 2022). To test whether our agents needed a longer training period to learn to generalize, we also trained base agents for 10,000 episodes (10x the original base agents).

During the imprinting trials, the artificial agents learned to detect and approach the familiar object (26 out of 30 agents). However, the artificial agents developed the same pattern of background-dependent recognition as the prior agents (Fig.3G).

**Pre-trained Encoders.** We also explored whether pre-trained encoders would help the artificial agents succeed at the task. First, we trained the encoders 'through the eyes' of newborn chicks, by simulating the first-person images acquired by chicks during early visual development (details in

Section A.7). Despite learning from stable (pre-trained) visual encoders, the artificial agents still developed background-dependent object recognition (Figure SI 3). Second, we inserted a variety of powerful pre-trained visual encoders into the agents, including Segment Anything (Kirillov et al., 2023), DINOv2 (Oquab et al., 2023), and Ego4D (Radosavovic et al. 2023; details in Section A.8). These pre-trained models perform well on standard computer vision benchmarks for object segmentation; nevertheless, artificial agents equipped with these encoders still failed to solve the one-shot learning task solved by newborn chicks (Figure SI 4). When we visualized the outputs of the pre-trained encoders, we found that some of the attention heads correctly segmented object features from background features. Thus, we speculate that the problem is not discovering useful visual features (pre-trained encoders appear to do this), but rather discovering which features to use to solve the *embodied* object recognition task.

**Animal-Machine Performance Gap.** To quantify the performance gap between the biological and artificial agents, we computed a "noise band" around the mean performance of the chicks (Fig.3B-G). The noise band reflects the average deviation of the chicks from mean chick performance. This defines a clear noise ceiling to judge prediction accuracy (Cao & Yamins, 2021). Noise ceilings capture the idea that the accuracy of an artificial agent in predicting chick behavior can only be as good as the accuracy of a chick in predicting chick behavior (e.g., due to measurement error and individual differences across chicks). If the average artificial agent performance reaches the noise band, then the artificial agent can be said to be "predictively adequate" of chick learning (i.e., the ML algorithms generate similar learning outcomes as the learning algorithms in chicks). This was not the case for this object segmentation NETT: there were large gaps in object recognition performance across the chicks and artificial agents.

## 4 DISCUSSION

We introduce a matched-experience Newborn Embodied Turing Test (NETT) for object segmentation. This NETT allows researchers to evaluate whether ML algorithms learn similar object segmentation abilities as newborn animals when provided with the same visual diet as animals. To create this NETT, we raised artificial agents in the same environments as newborn chicks from prior controlled-rearing experiments (Wood & Wood, 2021). We then tested whether the chicks and artificial agents learned common object recognition behavior. With impoverished training data (one object seen on one background), can ML algorithms match the rapid and robust learning of newborn chicks?

As a starting point, we inserted a variety of deep RL algorithms into the artificial agents and measured whether some algorithms can develop newborn-like object perception. We found remarkable consistency across algorithms. All of the algorithms successfully learned to imprint, but none of the algorithms learned *background-invariant* object representations that could generalize across novel backgrounds and viewpoints. Regardless of whether the artificial agents had small, medium, or large visual encoders, feedforward or recurrent policy networks, or dynamic processing in their visual sensors, the artificial agents all learned *background-dependent* representations, in which object features were incorrectly linked to background features. Unlike newborn chicks, modern deep RL algorithms, which largely rely on brute-force statistical association, are not capable of developing one-shot object segmentation in impoverished visual environments. These results show that there is a mismatch between the learning algorithms that drive biological versus artificial intelligence.

**NETTs: Developmentally Inspired Benchmarks for AI**. Until recently, attempts to build bridges between artificial and biological intelligence have focused almost entirely on benchmarks from mature humans and animals. While this approach has led to remarkable advances in AI, there is a growing consensus that building "naturally intelligent" learning algorithms will require taking development seriously. Biological intelligence develops across protracted periods of time, through nested periods of development (Smith et al., 2018; Adolph et al., 2018). Building animal-like intelligence may require allowing AI systems to develop under similar conditions. Even developmentally inspired AI approaches that train ANNs with the visual experiences of children (Orhan et al., 2020) or compare the behavior of ML models to human infants and children (Kosoy et al., 2020; Gandhi et al., 2021; Shu et al., 2021) still exclude critical learning experiences from the first months or years of life. NETTs uniquely capture the full environmental history of animals, allowing for rigorous evaluation of whether embodied ML algorithms learn like newborn animals.

**A Challenging One-Shot Learning Benchmark for AI**. Modern ML algorithms can solve object segmentation tasks, but only when trained on massive amounts of data (e.g., 14 million images from

ImageNet, Deng et al. 2009). When relying solely on brute-force statistical learning, ANNs need to see objects across many different contexts to successfully dissociate object and background features, and many still suffer from this "statistical concurrence problem" (Wood & Wood, 2021). For example, when ANNs are trained to recognize objects (e.g., dumbbells), but the objects are presented in similar scenes across training images (e.g., the dumbbells are always held by weightlifters), the resulting visual representations incorrectly bind object and background features (e.g., the dumbbell representations include hand and arm features; Mordvintsev et al. 2015).

To overcome this statistical concurrence problem, newborn brains use other mechanisms to segment objects from backgrounds. Specifically, motion-based segmentation plays a core role in the development of object perception. Young infants use motion cues to determine the 3-D shape of objects (Arterberry & Yonas, 2000; Owsley, 1983) and to integrate spatially separated parts into unified object concepts (Johnson et al., 2002; Kellman & Spelke, 1983; Kellman et al., 1986). Object segmentation through motion cues also develops *before* the ability to segment objects using static cues (e.g., color, shape; Spelke 1990). Likewise, for patients recovering from blindness, motion cues are essential for learning to segment and recognize objects (Ostrovsky et al., 2009). Motion-based segmentation thus appears to be a "primitive" of object perception—a program of visual learning that enables the brain to assemble fragmented features into unified object representations. Notably, motion-based segmentation allows learning systems to overcome the statistical concurrence problem, since motion can act as a selective-gating mechanism, constraining learning to moving features, rather than the whole visual field. Objects move separately from backgrounds, so motion-based segmentation can parse objects from backgrounds without the need for diverse visual experiences with objects.

We attempted to implement motion-based segmentation in artificial agents using LSTMs and dynamic visual sensors, but these components were clearly not sufficient to solve this one-shot segmentation task. Moving forward, we anticipate that our NETT will be a valuable benchmark for building embodied AI systems that can solve one-shot segmentation tasks, potentially through a brain-inspired motion-based segmentation process.

**Limitations**. One limitation of the current study is that we tested a limited number of ANNs. We hope that our NETT benchmark will allow AI researchers to plug a variety of learning algorithms into the artificial agents, in a community-wide effort to test whether any existing ML algorithms are capable of matching the rapid and robust visual learning of newborn chicks (Section A.9). A second limiting factor in building embodied models of biological intelligence is the availability of algorithms that can learn policies with high-dimensional action spaces. To avoid this roadblock, we built artificial agents with lower-dimensional action spaces. Future work might close this gap between animals and machines by building more realistic animal bodies, potentially through the use of sensorimotor pretraining (Radosavovic et al., 2023). A third limitation is that our artificial agents only had a single objective: to maximize their "imprinting reward." Real chicks, however, have multiple needs (e.g., hunger, thirst, temperature regulation, curiosity), which likely shape learning and behavior in important ways. In future studies, it would be interesting to build artificial agents with similar needs as biological chicks, potentially through the use of control systems that capture the homeostatic needs of biological systems (Yoshida et al., 2021) and/or curiosity-based rewards (Pathak et al., 2017).

**Broader Impact**. This project could lead to more powerful AI systems. Today's AI technologies are impressive, but their domain specificity and reliance on vast numbers of labeled examples are obvious limitations, especially compared to the rapid and flexible learning of newborn brains. NETTs will allow researchers to reverse engineer the core learning mechanisms in newborn brains, opening experimental paths towards building "naturally intelligent" ML systems.

## 5 ACKNOWLEDGEMENTS

We thank Bhargav Desai, Zachary Laborde, and Lalit Pandey for many helpful contributions to this project. All three deserve authorship based on contributions provided after the paper was submitted, but due to conference rules, we were not able to add new authors to the paper after submission. This project was funded by NSF CAREER grant (BCS-1351892, JNW), James McDonnell Foundation Understanding Human Cognition Scholar Award (JNW), and Facebook Artificial Intelligence Research award (JNW).

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

## A    Supplementary Material

### A.1    Comparing Avian and Mammalian Brains

How do avian brains compare to mammalian brains? Historically, scientists believed that avian brains were functionally different from and less sophisticated than mammalian brains (hence the pejorative term "bird brain"). Early studies of avian and mammalian brains were methodologically limited to identifying the general shape, appearance, and distribution of cells in brain regions (see Reiner 2005 for a thorough historical review). These studies broadly identified two major telencephalon components in mammalian brains: a layered "neocortex" and a radially organized "basal ganglia." In contrast, the avian telencephalon lacks layered regions, leading early scientists to propose that avian brains were dominated by a basal ganglia that operated largely through instinct rather than learning.

However, this view has been so thoroughly repudiated by the avian neuroscience community that the field completely revised the naming structure for the avian brain to reflect the deep structural and functional homologies between avian and mammalian brains (Reiner et al., 2004). Beginning in the 1960s, more advanced neuroanatomical methods revealed homologs for mammalian visual, auditory, somatosensory, and motor cortex in avian brains (Karten, 1969). Even though avian neocortex is organized radially rather than in layers, the avian neocortex shares connectivity patterns, physiological signatures, and cell types with mammalian neocortex (Stacho et al., 2020). Both mammalian and avian neocortex are organized in canonical circuits, which likely generate similar computational properties across taxa (Stacho et al., 2020).

Given the similarities in both the individual circuits and the large-scale brain organization across birds and mammals, we suspect that avian and mammalian brains are identical at an engineering-level of abstraction (i.e., the level at which we build machines and models that learn like animals). This hypothesis is further supported by a wealth of studies showing that some birds, like corvids and parrots, have forebrains of similar size as apes, live in complex social groups, and have long developmental periods before becoming independent (reviewed by Emery 2006). These birds show ape-like intelligence across a wide range of domains, in some cases even outperforming chimpanzees on cognitive tasks (Güntürkün & Bugnyar, 2016).

### A.2    Virtual Environments

The animal chambers had two 19" LCD monitors situated on opposite sides of the chamber: the LCD monitors were used to present the virtual objects to the chicks. The other two walls and ceiling were uniform white plastic. The floors of the chambers were constructed with wire mesh and had transparent troughs to hold food and water on one side of the chamber. The animal chambers measured 66 cm (length), 42 cm (width), and 69 cm (height).

To simulate the training and testing conditions of the chicks, we created virtual replicas of the animal chambers in the Unity Game Engine. We then trained and tested artificial agents in those chambers. The artificial agents had a height of 3.5 units and a length of 1.2 units. To monitor the agents' behavior, we positioned an invisible camera at the chamber's ceiling, capturing the top-view movement of the agent. The agent collected visual input (first-person RGB images, 64 x 64 resolution) through an invisible camera positioned in the agent's head as the agent moved through the chamber. Each image was passed through an encoder network before being passed to the policy network.

### A.3    Artificial Agents

#### A.3.1    Visual Encoders

We used small, medium, and large visual encoders (see Table 1). The "Small" encoder is a 3-layer convolutional network that outputs a 512-dimensional encoding vector. The "Medium" encoder is a 10-layer ResNet-10 model, incorporating an initial convolutional layer and subsequent ResBlocks that outputs a 128-dimensional encoding vector. The "Large" encoder follows a ResNet-18 architecture, incorporating an initial convolutional layer and four sequential layers of ResBlocks, resulting in 128-dimensional embedding vector.

#### A.3.2    Policy Networks

The base PPO Policy network uses the output of the feature extractor followed by a 1-layer MLP for the action and value networks to predict the optimal action. The recurrent policy network added

| Model | # Params | # Residual blocks | Form | Batch Size | Output Size |
|---|---|---|---|---|---|
| Small | 600,736 | None | CNN | 512 | 512 |
| Medium | 7M | 2 | CNN | 512 | 128 |
| Large | 12M | 8 | CNN | 512 | 128 |
| Dinov2(vit-s/14) | 21M | None | Transformer | 128 | 384 |
| SAM | 86M | None | Transformer | 128 | 256 |
| Ego4d | 21M | None | Transformer | 128 | 384 |
| SimCLR | 7M | 4 | CNN | 128 | 512 |
| Vanilla ViT | 16.4M | None | Transformer | 128 | 512 |

Table 1: Encoder networks

| Hyperparameter | Value |
|---|---|
| policy | CNN policy |
| n_steps | 2048 |
| learning_rate | 0.003 |
| n_epochs | 10 |
| ent_coef | 0.5 |
| gae_lambda | 0.95 |
| discount factor | 0.99 |
| clip_range | 0.2 |
| clip_range_vf | 0 |
| max_grad_norm | 0.5 |
| target_kl | 0.01 |

Table 2: Hyperparameter configuration for policy network

an additional 1-layer LSTM for the action and value networks in addition to the MLP layer. The hyperparameter configuration for the policy networks is described in Table 2.

## A.4 IMPRINTING REWARD

### A.4.1 ONE-EYED AGENT

Newborn chicks are naturally motivated to approach conspicuous, moving objects in their environment and form a social attachment to those objects. To mimic the internal, social reward of imprinting, we trained our chick agents using an "imprinting reward." The imprinting reward was designed to motivate the agents to spend time with the imprinting object by rewarding the agents for being close to the object and looking at the object.

The imprinting reward was based on the percentage of the agent's field of view (FOV) that was filled by the imprinting object. First, we computed the bounds of the object in the FOV. Next, we clamped the object bounds at the edge of the FOV so that only the visible parts of the object were included. Then, we used the X and Y coordinates of the object in the FOV to determine the maximum height (in y-coordinates) and width (in x-coordinates) the object covered in the FOV. Finally, we calculated the imprinting reward as object-height × object-width ÷ visual-field-area.

When the agent was close to the object and looking at it, the agent received a high imprinting reward. When the agent was far away from the object and looking at it, the agent received a small reward. When the agent was not looking at the object (i.e., the object was not in the FOV), the agent received no reward, regardless of how close it was to the object.

Critically, researchers using this benchmark in the future may "turn off" the imprinting reward in favor of using their own intrinsic reward. After all, newborn animals learn through intrinsic rewards. Building ML algorithms that learn like brains will thus require using biologically plausible intrinsic rewards, like curiosity, to mimic the learning capacities of brains.

### A.4.2 TWO-EYED AGENT

To compute the imprinting reward for the two-eyed agent, we calculate the imprinting reward for each eye (see One-eyed Agent section above). Then we compute the average of those two rewards as the two-eyed agent imprinting reward. For a visualization of the two-eyed agent, see SI Figure 1.

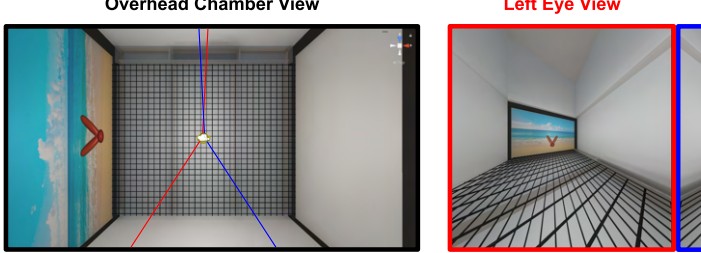

Supplementary Figure 1: The two-eyed version of the NETT. The agent is equipped with two eyes, one on each side of the head, with a small degree of overlap. The field of view of each eye is designed to match that of newborn chicks. **(Left)** The overhead view of the chamber, agent, and field of view of each eye (red and blue lines). **(Right)** The image input to the left (red) and right (blue) eyes. Researchers can concatenate the two images, stitch the images, use the images to compute depth, or whatever else might be useful for mimicking chicks' rapid and robust learning.

| Model | Familiar Object Familiar Background vs. Novel Object Novel Background | Familiar Object Novel Background vs. Novel Object Familiar Background | Familiar Object Familiar Background vs. Novel Object Familiar Background | Familiar Object Novel Background vs. Novel Object Novel Background |
|---|---|---|---|---|
| Small Encoder (PPO) | M = 72%
SD = 20%
t(28) = 5.79
p < 10-5
d = 1.08 | M = 31%
SD = 21%
t(29) = 5.03
p < 10-4
d = 0.92 | M = 50%
SD = 3%
t(27) = 0.28
p = .78
d = 0.05 | M = 52%
SD = 11%
t(29) = 0.97
p = .34
d = 0.18 |
| Medium Encoder (PPO) | M = 73%
SD = 20%
t(29) = 6.52
p < 10-6
d = 1.19 | M = 27%
SD = 20%
t(29) = 6.16
p < 10-5
d = 1.12 | M = 53%
SD = 10%
t(29) = 1.80
p = .08
d = 0.33 | M = 48%
SD = 11%
t(26) = 0.92
p = .37
d = 0.18 |
| Large Encoder (PPO) | M = 74%
SD = 20%
t(28) = 6.59
p < 10-6
d = 1.22 | M = 33%
SD = 20%
t(28) = 4.60
p < 10-4
d = 0.85 | M = 55%
SD = 12%
t(28) = 2.38
p = .02
d = 0.44 | M = 57%
SD = 17%
t(28) = 2.33
p = .03
d = 0.43 |
| DVS Encoder (PPO) | M = 69%
SD = 23%
t(29) = 4.38
p < 10-3
d = 0.80 | M = 31%
SD = 24%
t(29) = 4.33
p < 10-3
d = 0.79 | M = 51%
SD = 10%
t(29) = 0.42
p = .68
d = 0.08 | M = 50%
SD = 5%
t(29) = 0.29
p = .77
d = 0.05 |
| Small Encoder (LSTM) | M = 72%
SD = 20%
t(29) = 5.84
p < 10-5
d = 1.07 | M = 28%
SD = 20%
t(29) = 5.92
p < 10-5
d = 1.08 | M = 52%
SD = 6%
t(29) = 1.62
p = .12
d = 0.30 | M = 53%
SD = 14%
t(28) = 1.15
p = .26
d = 0.21 |
| 10k Training Episodes Small Encoder (PPO) | M = 70%
SD = 18%
t(29) = 6.01
p < 10-5
d = 1.10 | M = 35%
SD = 20%
t(29) = 4.07
p < 10-3
d = 0.74 | M = 52%
SD = 10%
t(29) = 1.22
p = .23
d = 0.22 | M = 51%
SD = 7%
t(29) = 1.07
p = .29
d = 0.20 |
| 10k Training Episodes Small Encoder (LSTM) | M = 64%
SD = 21%
t(29) = 3.77
p < 10-3
d = 0.69 | M = 34%
SD = 16%
t(29) = 5.16
p < 10-4
d = 0.94 | M = 51%
SD = 9%
t(29) = 0.48
p = .63
d = 0.09 | M = 48%
SD = 4%
t(29) = 2.53
p = .02
d = 0.46 |

Table 3: Results from models with visual encoders that learn alongside the policy network

## A.5    Training and Test Details

**Training Phase.** At the beginning of each training episode, the artificial chick was spawned at a random position and orientation within the chamber. Each training episode lasted 1,000 time steps. We trained the artificial agents for 1,000 episodes. For the Small Encoder, Medium Encoder, Large Encoder, DVS Encoder, and 10k Training Episodes agents, the encoder and policy networks were trained together during the Training Phase. After the Training Phase, the network weights were frozen for the Test Phase (i.e., the artificial agents did not receive any rewards during the Test Phase, and learning was discontinued).

**Test Phase.** Each artificial chick performed 1,080 object recognition trials (40 trials for each of the 27 background-viewpoint combinations presented to the chicks). At the beginning of each test episode, the artificial chick was spawned at the middle of the chamber, facing a blank wall. Each object recognition trial consisted of 1,000 time steps. Performance statistics are provided in SI Table 3.

## A.6    t-SNE Visualization of Visual Encoders

To visualize the internal working of the encoders, we generated t-SNE plots of the features encoded by the encoder networks. To create each visualization, we first created a dataset of images (first-person views from the agent) for each of the six backgrounds. We then extracted the encoder from the frozen model that was saved at the end of the training process. Finally, we ran the images through the encoder to generate feature embeddings. These embeddings were then visualized using t-SNE plots (shown in SI Figure 2).

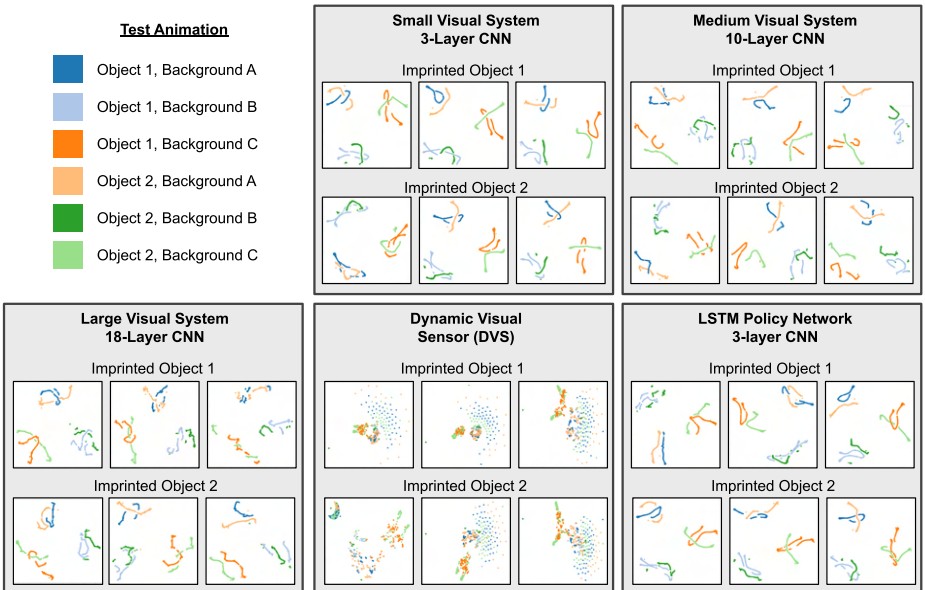

Supplementary Figure 2: t-SNE visualizations of the feature embeddings for images from the Test Phase. Each gray box shows a sample of t-SNE visualizations for a single ANN architecture. Top t-SNEs in each box are from agents imprinted to Object 1, and bottom t-SNEs in each box are from agents imprinted to Object 2. Each selected agent in each box also had a different imprinted background A (left), B (middle), or C (right). For all of the ANN architectures, the feature embeddings clustered based on the background rather than the object in the image.

## A.7    Training Visual Encoders 'Through the Eyes' of Newborn Chicks

To test whether artificial agents can solve this task when learning from stable visual features, we pre-trained visual encoders using simulated first-person images from inside the virtual controlled-rearing chamber, using the approach described by Lee et al. (2021a) and Pandey et al. (2023). To simulate the visual experiences of the chicks in Wood & Wood (2021), we simulated the visual inputs in the chick's environment by recording the first-person images acquired by an agent moving through the virtual chambers (SI Figure 3). The agent could move forward and backward, turn left and right, and

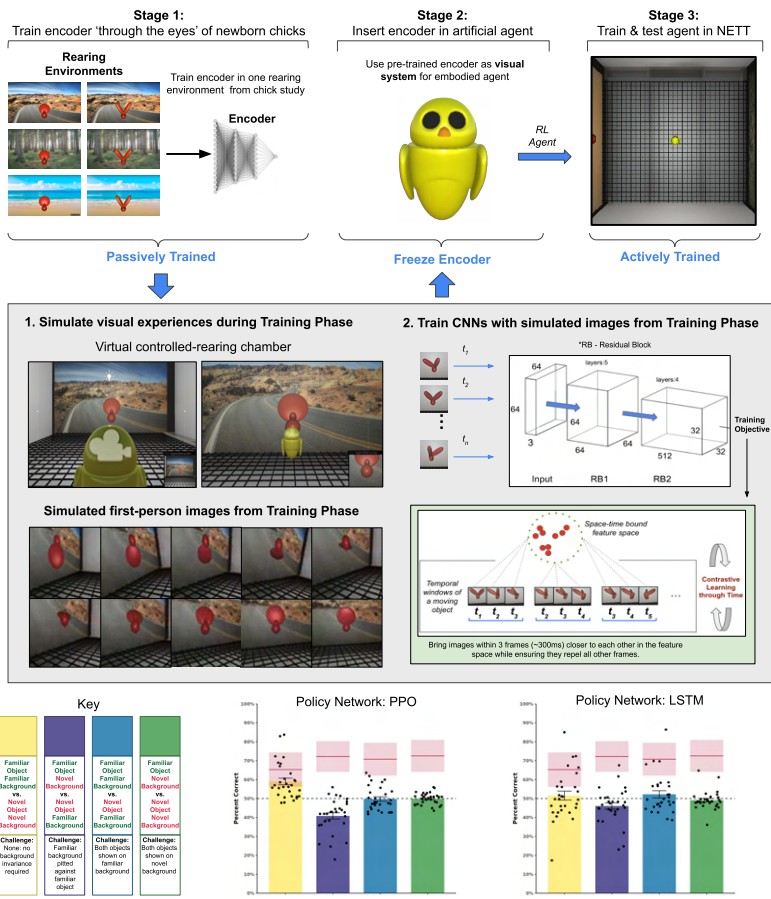

Supplementary Figure 3: Training visual encoders 'through the eyes' of newborn chicks. In Stage 1, we simulated the visual experiences of chicks during the training phase, then trained CNN encoders with those simulated images using a CLTT learning objective. In Stage 2, we froze the encoders and used them as visual systems for embodied agents. In Stage 3, we trained and tested the agents (with frozen encoders, but learning policy networks) in the NETT task. Bottom panel shows performance.

rotate its head along the three axes of rotation (yaw, pitch, roll). We collected 80,000 first-person images for each of the six rearing conditions and used those images to train the encoders.

For the encoders, we used a CNN (ResNet) backbone with 10 layers. We then trained the CNN by exploiting the temporal structure of natural visual experience. There is ample evidence that newborn animals (including chicks) use time as a teaching signal, suggesting that unsupervised temporal learning plays a foundational role in the earliest stages of visual learning (Wood, 2016; Wood & Wood, 2016; 2018; 2021; Matteucci & Zoccolan, 2020). To mimic early temporal learning, we used a contrastive learning objective function that leverages the temporal structure of visual experience, without relying on arbitrary image augmentations or labels (Schneider et al., 2021). The algorithm—Contrastive Learning Through Time (CLTT)—contrasts temporally adjacent instances (positive examples) against randomly selected non-adjacent instances (negative examples), thereby learning representations that capture the underlying dynamics, context, and patterns across time (SI Figure 3). After training the encoders, we inserted them into the artificial agents, then trained and tested the agents in the NETT. The encoders were frozen during the RL training.

The encoders were pre-trained in the same environment in which we trained the artificial agents. We reasoned that training the encoder and policy networks sequentially, rather than simultaneously, might help the agents solve the task, since the frozen encoders would provide stable visual features for learning the embodied object recognition task (as demonstrated by Parisi et al. 2022). However, like all other agents we tested, the agents failed to solve the task, largely learning background-dependent object representations (SI Figure 3).

| Model | Familiar Object Familiar Background vs. Novel Object Novel Background | Familiar Object Novel Background vs. Novel Object Familiar Background | Familiar Object Familiar Background vs. Novel Object Familiar Background | Familiar Object Novel Background vs. Novel Object Novel Background |
|---|---|---|---|---|
| Pretrained SimCLR (PPO) | M = 59% | M = 41% | M = 50% | M = 50% |
| | SD = 10% | SD = 9% | SD = 6% | SD = 3% |
| | t(29) = 5.19 | t(29) = 5.64 | t(29) = 0.22 | t(29) = 0.23 |
| | p < 10-4 | p < 10-5 | p = .83 | p = .82 |
| | d = 0.95 | d = 1.03 | d = 0.04 | d = 0.04 |
| Pretrained SimCLR (LSTM) | M = 52% | M = 46% | M = 52% | M = 49% |
| | SD = 13% | SD = 9% | SD = 10% | SD = 5% |
| | t(29) = 0.67 | t(29) = 2.35 | t(29) = 1.18 | t(29) = 0.94 |
| | p = .51 | p = .03 | p = .25 | p = .36 |
| | d = 0.12 | d = 0.43 | d = 0.21 | d = 0.17 |

Table 4: Results of artificial agents with encoders pre-trained 'through the eyes' of newborn chicks

## A.8 INSERTING POWERFUL PRE-TRAINED ENCODERS INTO ARTIFICIAL AGENTS

We also tried inserting a variety of powerful pre-trained visual encoders into the agents, including DINOv2 (Oquab et al., 2023), Ego4D (Radosavovic et al., 2023), and Segment Anything (Kirillov et al., 2023). The pre-trained encoders were trained on millions of images/videos capturing diverse real-world experiences, far exceeding the sparse visual experiences of the chicks. This approach thus violates the spirit of a *matched-experience* NETT, which involved matching the visual experiences acquired by newborn animals and machines. Nevertheless, this approach can provide valuable insight into *why* artificial agents fail at this task. We found that agents equipped with powerful pre-trained encoders still fail to solve the one-shot learning task solved by chicks. While these pre-trained models perform well on computer vision benchmarks for object segmentation, they are not sufficient to solve the one-shot learning task solved by chicks (SI Figure 4). When we visualized the outputs of the pre-trained encoders, we found that some of the attention heads correctly segmented object features from background features. Thus, we speculate that the problem is not discovering useful visual features (pre-trained encoders appear to do this), but rather discovering which features to use to solve the *embodied* object recognition task.

**Pre-trained Encoders.** We used the following pre-trained encoders (SI Table 1):

1. Dinov2 (Oquab et al., 2023) is a self-supervised learning technique applied on Vision Transformers developed by Meta Research. The model enables all-purpose visual features i.e., features that work across image distributions and tasks without finetuning.

2. Segment Anything Model (SAM) (Kirillov et al., 2023) is a foundation model trained on over 1 billion annotations, predominantly for natural images, that is intended to segment user-defined objects of interest in an interactive manner. The model was developed and released by Meta Research.

3. Ego4D (Radosavovic et al., 2023) is a pretrained masked autoencoder (MAE) model with a ViT-S architecture trained on the Ego-Soup dataset, which consists of Ego4D and other egocentric human video datasets.

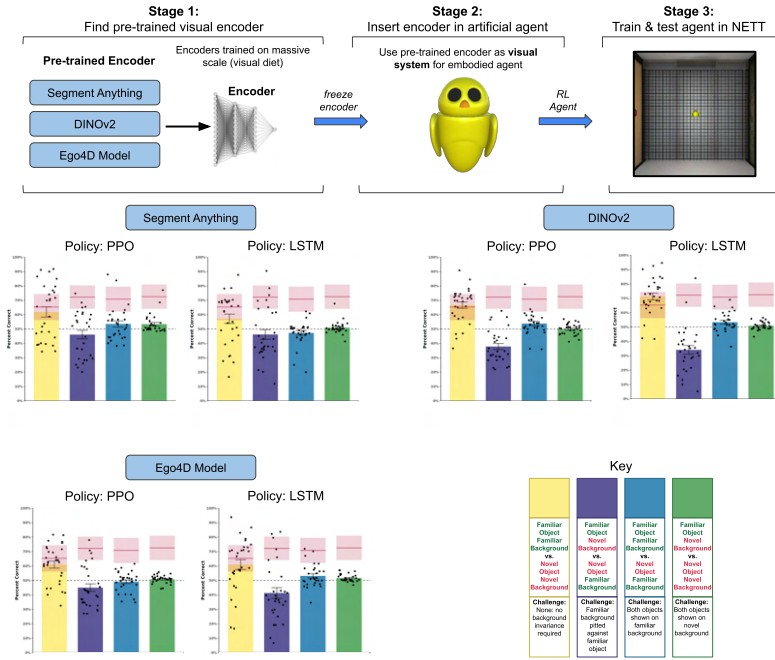

Supplementary Figure 4: Schematic of artificial agents with pre-trained visual encoders. In Stage 1, selected a set of pre-trained encoders that are noted for their segmentation abilities. In Stage 2, we inserted the pre-trained encoders as visual systems for embodied agents. In Stage 3, we trained and tested these agents (with frozen encoders, but learning policy networks) in the NETT task. Bottom panel shows performance.

| Model | Familiar Object Familiar Background vs. Novel Object Novel Background | Familiar Object Novel Background vs. Novel Object Familiar Background | Familiar Object Familiar Background vs. Novel Object Familiar Background | Familiar Object Novel Background vs. Novel Object Novel Background |
|---|---|---|---|---|
| Pretrained SAM (PPO) | M = 62% | M = 46% | M = 53% | M = 53% |
| | SD = 19% | SD = 16% | SD = 12% | SD = 6% |
| | t(29) = 3.32 | t(28) = 1.30 | t(28) = 1.59 | t(28) = 2.86 |
| | p = .002 | p = .21 | p = .12 | p = .01 |
| | d = 0.62 | d = 0.24 | d = 0.29 | d = 0.53 |
| Pretrained SAM (LSTM) | M = 57% | M = 46% | M = 47% | M = 51% |
| | SD = 18% | SD = 19% | SD = 9% | SD = 4% |
| | t(29) = 2.19 | t(29) = 1.12 | t(29) = 1.71 | t(29) = 1.21 |
| | p = .04 | p = .27 | p = .10 | p = .24 |
| | d = 0.40 | d = 0.20 | d = 0.31 | d = 0.22 |
| Pretrained DINOv2 (PPO) | M = 67% | M = 38% | M = 54% | M = 50% |
| | SD = 12% | SD = 11% | SD = 8% | SD = 4% |
| | t(29) = 7.49 | t(29) = 5.83 | t(29) = 2.57 | t(29) = 0.19 |
| | $p < 10^{-7}$ | $p < 10^{-5}$ | p = .02 | p = .85 |
| | d = 1.37 | d = 1.06 | d = 0.47 | d = 0.03 |
| Pretrained DINOv2 (LSTM) | M = 71% | M = 34% | M = 53% | M = 51% |
| | SD = 13% | SD = 16% | SD = 7% | SD = 3% |
| | t(28) = 8.83 | t(29) = 5.31 | t(28) = 2.67 | t(28) = 1.64 |
| | $p < 10^{-8}$ | $p < 10^{-4}$ | p = .01 | p = .11 |
| | d = 1.64 | d = 0.99 | d = 0.50 | d = 0.31 |
| Pretrained Ego4d (PPO) | M = 61% | M = 45% | M = 49% | M = 51% |
| | SD = 13% | SD = 14% | SD = 7% | SD = 3% |
| | t(29) = 4.51 | t(29) = 1.92 | t(29) = 0.72 | t(29) = 1.67 |
| | $p < 10^{-4}$ | p = .07 | p = .48 | p = .11 |
| | d = 0.82 | d = 0.35 | d = 0.13 | d = 0.31 |
| Pretrained Ego4d (LSTM) | M = 61% | M = 41% | M = 53% | M = 52% |
| | SD = 18% | SD = 20% | SD = 8% | SD = 3% |
| | t(29) = 3.26 | t(29) = 2.35 | t(29) = 2.18 | t(29) = 3.57 |
| | p = .003 | p = .03 | p = .04 | p = .001 |
| | d = 0.59 | d = 0.43 | d = 0.40 | d = 0.65 |

Table 5: Results from models with encoders were pretrained on a massive visual diet

### A.9 NETT BENCHMARK FLEXIBILITY

We emphasize that researchers can flexibly control how artificial agents are trained in the NETT. Our NETT runs with Stable Baseline 3 (Raffin et al., 2021), so researchers can vary (among other parameters) the number of training episodes, length of the training episodes, batch size, number of test episodes, length of test episodes, learning rate, architecture of visual encoders, architecture of policy networks, and the nature of the intrinsic reward(s). Users can also vary the image resolution of the eyes (cameras) and whether the agent has one eye or two eyes (SI Figure 1). Finally, to help visualize and study machine behavior through the same lens as animal behavior, the NETT allows users to record images from an overhead camera and from a camera capturing the agent's first-person views.

### A.10 DATA AVAILABILITY

The object segmentation NETT reported in this paper can be accessed at the following link: https://github.com/buildingamind/nett-object-segmentation For additional NETTs and details on running this and other NETTs, see: https://origins.luddy.indiana.edu

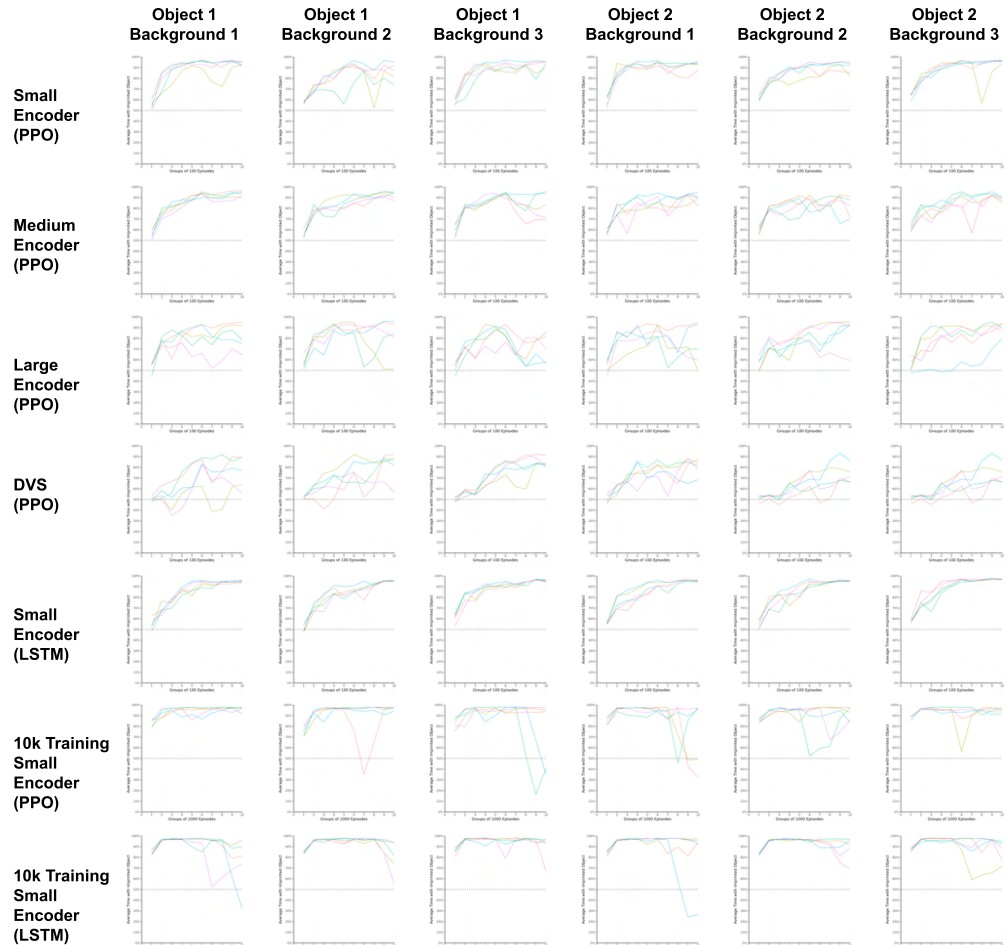

Supplementary Figure 5: Performance across the training phase for agents whose visual encoders learn along with the policy network.

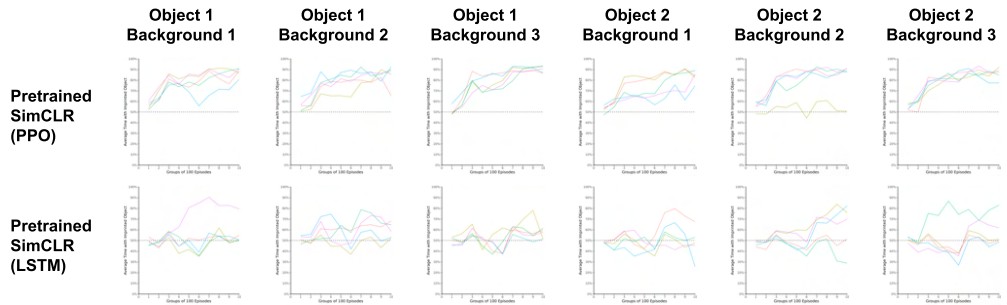

Supplementary Figure 6: Performance across the training phase for agents whose visual encoders were pre-trained with simulated visual input from the digital chamber.

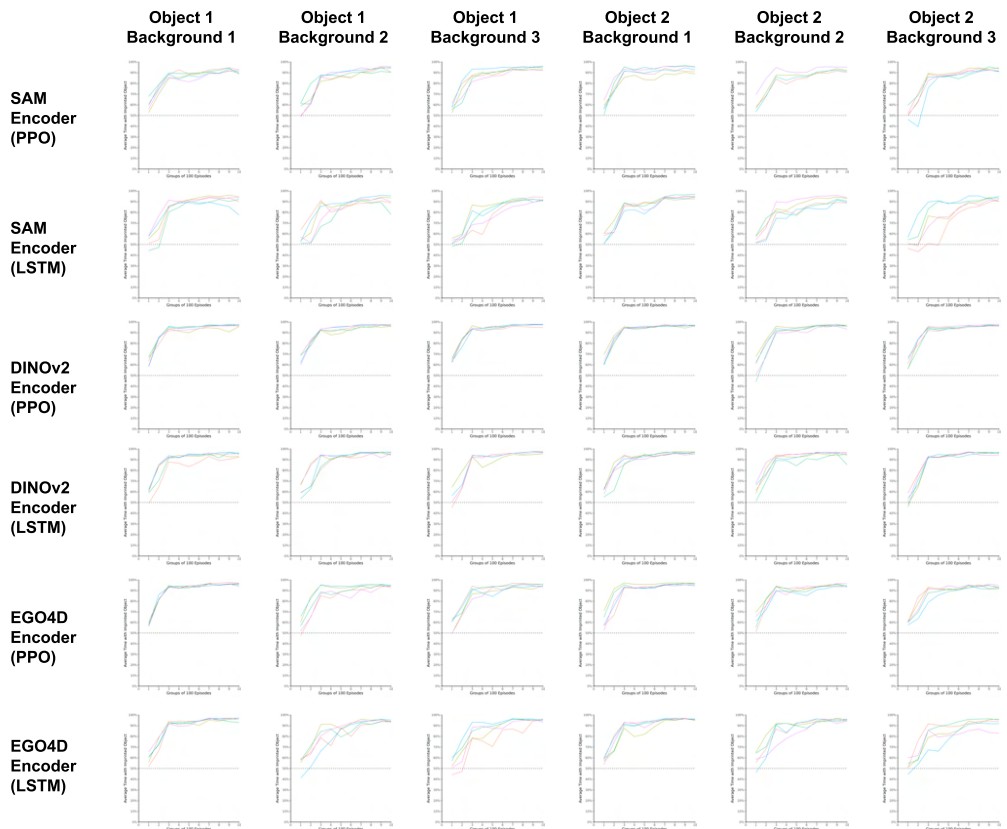

Supplementary Figure 7: Performance across the training phase for agents whose visual encoders were pretrained on a large and varied visual diet.

