# OpenReview forum: "A Newborn Embodied Turing Test for Comparing Object Segmentation Across Animals and Machines"
_ICLR.cc/2024/Conference — ICLR 2024 poster_

### Official Review · Reviewer_ELqk · 2023-10-28

**Soundness:** 2 fair
**Presentation:** 2 fair
**Contribution:** 2 fair
**Rating:** 6
**Confidence:** 4

**Summary:**

This work introduces a benchmark, the Newborn Embodied Turing Test (NETT), and reveals the limitations of current machine learning algorithms compared to newborn brains in object segmentation tasks. However, the experimental results of this work are insufficient to validate the claimed fairness of the NETT.

**Strengths:**

1.	This work's main contribution is the introduction of the Newborn Embodied Turing Test (NETT) benchmark, which enables a direct comparison of the learning abilities between newborn animals and machines in object segmentation tasks.

**Weaknesses:**

1.	The evaluation of this proposed NETT benchmark is not comprehensive enough. Many variables were not taken into consideration during the data collection process for the biological chicks, such as their binocular vision compared to the single camera of artificial intelligence chicks. Meanwhile, biological chicks might roam around while artificial intelligence chicks do not. These unaccounted variables raise doubts about the effectiveness of the proposed benchmark. The validity of the benchmark may be compromised due to the discrepancies in sensory capabilities and behavior between biological and artificial chicks. Further refinement and consideration of these variables are necessary to ensure the reliability and validity of the benchmark.
2.	The presentation of this work still needs improvement, as the figures are not in vector format. For example, Figure 3 lacks clear labels for the x and y axes, making it difficult to understand the meaning of the figure.

**Questions:**

1.	The evaluation of this proposed NETT benchmark is not comprehensive enough. Many variables were not taken into consideration during the data collection process for the biological chicks, such as their binocular vision compared to the single camera of artificial intelligence chicks. Meanwhile, biological chicks might roam around while artificial intelligence chicks do not. These unaccounted variables raise doubts about the effectiveness of the proposed benchmark. The validity of the benchmark may be compromised due to the discrepancies in sensory capabilities and behavior between biological and artificial chicks. Further refinement and consideration of these variables are necessary to ensure the reliability and validity of the benchmark.
2.	The presentation of this work still needs improvement, as the figures are not in vector format. For example, Figure 3 lacks clear labels for the x and y axes, making it difficult to understand the meaning of the figure.

---

> ### Author Response · Authors · 2023-11-22
>
> Thank you for the constructive feedback.
>
> **The Reviewer suggests that our NETT is not comprehensive enough because biological chicks have two eyes, whereas the artificial chicks had one eye.**
>
> The Reviewer correctly points out that we did not fully close the experience gap between newborn chicks and artificial agents. Newborn chicks have two eyes, whereas our artificial agents had one eye. This difference may have led to the divergent learning outcomes in the biological and artificial agents. To allow researchers to explore this possibility, we created a “two-eyed agent” (Fig. S4). The two-eyed agent was carefully constructed so that the FOV of each eye matches the FOV of newborn chicks.
>
> There are many ways in which researchers might process the two eye images in the visual encoder, so we modified our benchmark so researchers can select either a “one-eyed” or “two-eyed” agent. For the “two-eyed” agent, the benchmark will output two images on each time step, one from each of the two cameras, which are positioned on each side of the agent’s head (akin to the placement of eyes on chicks). Researchers can then build and test their own binocular artificial agents using the revised NETT, allowing for systematic exploration of the computational role of one vs. two eyes on the development of object segmentation.
>
> **The Reviewer suggests another difference between biological and artificial systems is that the biological chicks might roam around while the artificial chicks do not.**
>
> To clarify, our PPO and LSTM PPO policies have an entropy parameter governing how much the agents move at random (akin to roaming). For all of our models, the entropy parameter starts high and linearly decays throughout training. This parameter encourages the agents to explore early in learning.
>
> That said, we agree that different types of rewards might promote more biologically inspired roaming and information-seeking behavior. We informally tried a number of different rewards and found that this imprinting reward produced behavior that best matched the imprinting behavior of newborn chicks. However, newborn animals learn through intrinsic rewards, so building ML systems that learn like brains will require ‘turning off’ our imprinting reward and adding a more biologically plausible intrinsic reward to the agents.
>
> In our revision, we now note that we designed the benchmark so researchers can use their own custom-designed intrinsic rewards to drive the agent’s behavior (i.e., researchers do not need to use our imprinting reward). This makes the NETT more flexible, allowing researchers to plug any encoder, policy network, and/or intrinsic reward into the artificial agents. We hope researchers will use this benchmark to discover reward structures that produce animal-like object perception in ML systems, perhaps by encouraging the artificial agents to more actively explore their environment, as the Reviewer suggests.
>
> **The Reviewer notes that the figures need improvement.**
>
> As suggested, we improved the quality of the figures and added clear axis titles.

---

> ### Comment · Reviewer_ELqk · 2023-11-23
>
> Thanks for author's answer. Although the gap between biological chicks and AI chicks is still doubtful, the authors' rebuttle have solved most of my concerns and more detailed experimental results have been given. Hence, I decided to raise my rate to 6.

---

### Official Review · Reviewer_FxD2 · 2023-10-30

**Soundness:** 4 excellent
**Presentation:** 3 good
**Contribution:** 4 excellent
**Rating:** 8
**Confidence:** 5

**Summary:**

The authors aimed to put both state-of-art vision models and biological vision on similar levels of data diet. They do this by rearing newborn chicks in a very controlled visual environment where they have access to limited visual experience. By exploiting the imprinting ability of chicks, their ability to segment foreground from background is studied. Then vision models are trained with exact same data by instantiating them as the vision modules in a RL agent rewarded to go towards the imprinting object. Finally both the models and chicks are probed for their ability to segment new objects and/or new backgrounds and it was found that the models fare much worse compared to the newborn chicks. This indicates that the state-of-art models lack something that newborn chicks' visual system has.

**Strengths:**

Rarely do I see works with this novelty, quality and significance.

* Studying deep learning models in a setting where the data diet is controlled is timely and needed.
* The data collected from newborn chicks is compelling and novel.
* The results showing that current models lack the necessary mechanisms can be very impactful.
* The data collection procedure and analysis opens up a ton of possibilities for future work.
* The paper is overall well written.

**Weaknesses:**

* Perhaps the biggest weakness is that the vision models here are trained in a way that is not very indicative of the state-of-the-art. Most vision models that is used in the community are usually a combination of self-supervised training and supervision with a lot of exemplars. While this might diminish how relevant this work is for the community, it does not take away from the significance of the work since the goal of the work is to study models when they get only as much visual experience as newborn chicks (i.e newborn chicks are not getting a ton of supervision).
* PPO agents are trained for a lot of episodes (original PPO paper (Schulman, 2017) did 1M episodes) usually while this work only does 1000 episodes. This is probably okay since this setting is much simpler. But I would still train one model for much more steps to rule out strange behaviors (like grokking (Power, 2022))
* The paper might benefit from an expanded section on the differences between birds and mammal brains, especially since the audience is likely to be more from the computational side. While I do appreciate the effort in section 1.1, it might be helpful to expand it a bit more.

**Questions:**

* Why does Figure 1 make it seem like this is a cyclic process? I don't see why it is put on a circle - step 1 does not follow step 6?
* I wish this was called something other than a Turing test. I think most people (as far as I know) think of the imitation game when they hear Turing test while this is very different. I feel like this is significantly different enough to warrant a different. I almost feel like calling this Turing test is underselling it. Would another name that highlights the "limited exposure to environment" aspect of this work be better suited? Turing test says nothing about how much experience the models get to have?
* Section 2.1 : Was the second display blank (all black) or just a background?
* Section 2.2 : Do the RL agents need to be called "artificial chicks"? I think calling them "RL agents" or something is less confusing.



Improvement to the paper -
* All the figures need major revamp. The text is barely readable and they need to be of higher resolution. Figure 2 needs to say what "F" and "N" stand for. Figure 2 could also be made such that the backgrounds and foregrounds are easily distinguishable.

---

> ### Author Response · Authors · 2023-11-22
>
> Thank you for your feedback! We will keep your words of encouragement close at hand as we forge ahead with this research program.
>
> **The Reviewer argues that the biggest weakness of the paper is that we did not use state-of-the-art systems in object segmentation. The reviewer notes that this contradicts the goals of the benchmark (which requires matching the training data across animals and machines), but nevertheless convincingly argues that this would help integrate the paper into the ICLR community.**
>
> We agree and implemented the suggestion in two ways:
>
> First, we inserted a variety of powerful pre-trained visual encoders into the agents, including DINOv2 (Oquab et al., 2023), Ego4D (Radosavoic et al., 2022), and Segment Anything (Kirillov et al., 2023). These pre-trained models perform well on standard computer vision benchmarks for object segmentation; nevertheless, artificial agents equipped with these pretrained encoders still failed to solve the one-shot learning task solved by newborn chicks. As shown in Fig. S3, agents with pre-trained encoders learned background-dependent object representations.
>
> Second, we pre-trained the visual encoders using simulated first-person images from inside the controlled-rearing chambers, mimicking the visual experiences of the chicks (Fig. S2). The encoders were thus pre-trained in the same environment in which we tested the artificial agent. We then inserted those frozen pre-trained encoders into the artificial agents, allowing the policy network to learn from a stable visual encoder. We reasoned that training the encoder and policy networks sequentially, rather than simultaneously, might help the agents solve the task, since the frozen encoders would provide more robust visual experiences for learning the embodied imprinting task. However, like all the other agents we tested, the agents failed to solve the task, largely learning background-dependent object representations.
>
> **The Reviewer notes that since we only trained for 1K episodes, it is possible that more powerful generalization abilities might emerge with more training.**
>
> Great suggestion. To test this possibility, we trained the artificial agents with 10X more episodes (10,000 episodes). As shown in Fig. 3G, the agents still learned background-dependent object representations, akin to agents trained for 1,000 episodes. We also emphasize in the revision that we chose the 1,000 training episodes after observing that the artificial agents successfully learned to imprint in 1,000 episodes (as shown in Fig. S5).
>
> **The Reviewer wonders why we visualized the NETT as a circular process.**
>
> In the Figure 1 caption, we now explain that we see this NETT as the foundation for a closed-loop scientific paradigm, in which researchers build ML models that learn like newborn animals, and then use those models to generate predictions for new experiments with newborn animals to validate and refine those models.
>
> **The Reviewer wishes we named this something other than a Turing test, since Turing tests don’t involve matching the experiments across biological and artificial systems.**
>
> We appreciate the feedback and agree with the sentiment. After reading this comment, our team spent a long time iterating on names. Ultimately, there were 4 elements we wanted to emphasize: (1) matching the experience between biological and artificial systems, (2) focusing on the origins of intelligence, (3) the fact that the models are embodied agents, rather than disembodied visual systems, and (4) comparing intelligence between animals and machines. We’ve settled on referring to the benchmark as a matched-experience Newborn Embodied Turing Test (NETT).
>
> **The Reviewer suggests adding an expanded section comparing bird and mammalian brains.**
>
> Great suggestion. We added a more detailed discussion of the differences between avian and mammalian brains to the Supplementary Methods (Section A.1).
>
> **The Reviewer suggests using the term “artificial agents” rather than “artificial chicks.”**
>
> We agree and have changed the manuscript accordingly.
>
> **The Reviewer suggests clarifying the stimuli presented on the 2nd display walls and Fig. 3 labels.**
>
> We revised the manuscript accordingly to indicate that the 2nd display wall was a white screen. We also improved the quality of the figures and clarified what “F” and “N” stand for in the figures.

---

### Official Review · Reviewer_oTEc · 2023-11-03

**Soundness:** 3 good
**Presentation:** 3 good
**Contribution:** 3 good
**Rating:** 8
**Confidence:** 4

**Summary:**

## Update

I thank the authors for the answer. I stand by initial review and believe this paper should be accepted.

## Original

This paper aims to benchmark the intelligence of embodied AI systems against that of biological ones (animals). Towards this end, they propose a new benchmark, the Newborn Embodied Turing Test (NETT).

In NETT, an agent is born into an environment with a display showing a singular rotating object. The agent must then learn to properly identify that object when presented with distractor objects or when it is placed on novel backgrounds. The biological agents are chicks that learn this skill via filial imprinting and the artificial agents are DRL agents that learn this via an filial-imprinting-like reward.

The authors find that while biological chicks are well suited to learn this task, their artificial counterparts are not. The authors consider a variety of DNN architectures and find that none perform significantly above chance.

**Strengths:**

This paper presents a convincing experiment that directly compares the capabilities of biological and artificial chicks. I really like how controlled the setup is. You can't perfectly control for all the differences between biological and artificial systems (as the authors note in their limitation section), but this paper does an admirable job at reducing this gap as much as possible.

The authors examine a variety of different DNN architectures with the aim to close the gap between biological and artificial chicks.

The discussion section is well-written and presents a balanced discussion that points out the limitation of the experiments presented and contains interesting pieces of information.

**Weaknesses:**

Figures in the paper are very low resolution (and possibly heavily compressed with lossy compression). This is particularly noticeable for figure 3, which is nearly illegible.

The number of trials used for training artificial chicks seems rather small at 1000. Similarly, the episode length of 1000 seems long given how small the environment is. Why were these numbers chosen? Are they similar to the number of times the biological chicks turn towards their object and how long they stay facing it?

There has been work in deep reinforcement learning algorithms that may be applicable here. One example is as DAAC and IDAAC ("Decoupling Value and Policy for Generalization in Reinforcement Learning", Raileanu and Fergus).

**Questions:**

### Questions

1. Did the authors perform the biological chick experiments themselves or are those directly from Woods & Woods 2021? (I was unable to get access to this paper to check myself.) If the authors performed this experiment, has it been checked by an Institutional Review Board (IRB) or similar?

### Suggestions for Improvement

This paper reminded me of "Exploring Exploration: Comparing Children with RL Agents in Unified Environments" by Kosoy et al. These two works perform different experiments and have different goals, but the authors may want to mention this work.

Some "high-water" marks for artificial chicks would be useful. If one was trained on the test set, how would it do? Ideally it should perfectly solve the task. If one was trained with a singular object but many different backgrounds, how would it do?

**Details Of Ethics Concerns:**

~This paper contains experiments performed with real animals in not great conditions -- newborn chicks are confided to a small box in isolation with only a rotating object on a screen for company. While the experimental design seems to come from prior work (Woods & Woods, "One-shot object parsing in newborn chicks", 2021), I don't feel confident making a judgement on if this experiment is okay.~

~I asked the authors if they performed this experiment and if they did, has it been check by an IRB or similar.~

The authors did not perform the experiment with artificial chicks.

---

> ### Author Response · Authors · 2023-11-22
>
> Thank you: we appreciate your time and feedback.
>
> **The Reviewer notes the low resolution figures.**
>
> We improved the resolution quality of the figures.
>
> **The Reviewer notes that since we only trained for 1K episodes, it is possible that more powerful generalization abilities might emerge with more training.**
>
> Great suggestion. To test this possibility, we trained the artificial agents with 10X more episodes (10,000 episodes). As shown in Fig. 3G, the agents still learned background-dependent object representations, akin to agents trained for 1,000 episodes. We also emphasize in the revision that we chose the 1,000 training episodes after observing that (1) the artificial agents successfully learned to imprint in 1,000 episodes, and (2) the artificial agents developed a robust form of object recognition in 1,000 episodes.
>
> **The Reviewer also suggests a related paper by Kosoy et al. and points us toward new work in deep reinforcement learning models (DAAC and IDAAC) that might be applicable.**
>
> Our revision now cites the Kosoy et al. paper. We unfortunately were not able to test DAAC and IDAAC within this rebuttal period, but we will add this to our list of models to try next. We also note in the revision that we designed our benchmark to be as easy to use as possible, so we hope other researchers will try DAAC, IDAAC, and any other candidate models that might be interesting to researchers across artificial intelligence and the mind sciences.
>
> **The Reviewer wonders whether we performed the chick experiments ourselves to check that necessary animal welfare/ethics procedures were followed.**
>
> The chick experiments were conducted by Wood & Wood (2021) and not conducted for this paper. We have revised the text to make this clearer for readers. Our paper extends this prior work by turning the behavioral results into a NETT benchmark. The original paper (Wood & Wood, 2021) specifies that the work was approved by an IACUC (pg. 2411).
>
> **The Reviewer suggests adding “high-water” marks for the artificial chicks, by training and testing them on the test set.**
>
> Great idea. While we did not have time to add “high-water” makers to this revision, we plan to add them to the camera-ready version if the paper is accepted.

---

### Official Review · Reviewer_gdnS · 2023-11-05

**Soundness:** 1 poor
**Presentation:** 2 fair
**Contribution:** 2 fair
**Rating:** 3
**Confidence:** 4

**Summary:**

This paper conducts a comparison study on the one-shot object segmentation ability of real animals and machines. Specifically,  the authors create simulated ‘digital twin’ environments that mimic the rearing condition of real biological newborn chicks. In the simulated environments, the ‘artificial chick’ is trained via deep reinforcement learning to segment objects. The experimental results show that ‘artificial chick’ failed in the one-shot object segmentation task while biological chicks are able to solve it.

**Strengths:**

Originality and Significance:
The reviewer found the idea of comparing machine and real animals' learning ability in a strictly controlled environment interesting. This line of study might help us learn more about the similarity and difference of animal brain and deep neural network.


Quality:
While the idea of the paper is interesting, the reviewer found some claims are not well-supported by the experiments. Please see the Weakness section for more details.


Clarity:
The presentation is mostly clear. However, many improvements, particularly on the figures, are needed.  Please see the Weakness section for more details.

**Weaknesses:**

1. The technical contribution of the paper is limited. Specifically, the proposed ‘artificial chick’ is a PPO agent trained with ‘imprinting reward’, which encourages the agent to move close to an object. It is not surprising that this simple baseline with a heuristic reward fails to solve the one-shot object segmentation task.

2. The reviewer found main claims in the paper are not well-supported by the experiments. Specifically, the authors claimed that ‘’... none of the algorithms learned background-invariant object representations that could generalize across novel backgrounds and viewpoints”. However, the ML algorithm the authors used in the paper are just  PPO with different architectures and a heuristic reward. The reviewer thinks those simple methods well-represent the state-of-the-art one-shot object segmentation methods. There are many works on one-shot segmentations [1, 2]. Incorporating those methods in the ‘artificial chick’ might make the experiments more convincing.

3.  The reviewer has concerns on the significance of the paper. The main finding here is that the RL trained agent failed the one-shot segmentation task. It seems expected. Providing more insights and analysis of why it fails may make the paper more valuable.

4.  The presentation in the experimental section is unclear. Particularly, the text and numbers in figure 3, which show the only experimental result of the paper, are ineligible.



[1] Learning to Segment Rigid Motions from Two Frames, Yang et al., CVPR 2021.
[2] One-Shot Learning for Semantic Segmentation, Shaban et al., BMVC 2017

**Questions:**

1. According to section 2, the biological chicken data is from previous work (Wood et al. 2021). However, in the abstract, the authors claim that “we raised newborn chicks in controlled environments …”. This seems contradicting. Could you clarify?


2.  In Introduction, it reads “However, most studies with newborn subjects have produced data with a low signal-to-noise ratio,”. The reviewer has difficulty following the content. Please give citations for ‘some studies’ and elaborate what is the ‘produced data’ here.

---

> ### Author Response · Authors · 2023-11-22
>
> Thank you for the helpful feedback.
>
> **The Reviewer argues that the claims are not well-supported by the experiments because we used a PPO agent trained on a heuristic (imprinting) reward. The Reviewer suggests adding state-of-the-art object segmentation models into the artificial agents to make the experiments more convincing.**
>
> This is a great idea. We implemented this suggestion in three ways:
>
> First, we took three top-performing object segmentation models—DINOv2 (Oquab et al., 2023), Ego4D (Radosavoic et al., 2022), and Segment Anything (Kirillov et al., 2023)—and inserted them into the artificial agents. The pre-trained encoders were trained on millions of images across one thousand object categories, far exceeding the sparse visual experiences of the chicks. Nevertheless, artificial agents equipped with these pre-trained encoders still failed to solve the one-shot learning task solved by newborn chicks. As shown in Fig. S3, agents with pre-trained encoders learned background-dependent object representations.
>
> Second, to allow the policy network to learn from a stable visual encoder, we pre-trained visual encoders using simulated first-person images from inside the controlled-rearing chambers (Fig. S2). The encoders were thus pre-trained in the same environment in which we trained the artificial agents, mimicking the visual experiences of the chicks. We reasoned that training the encoder and policy networks sequentially, rather than simultaneously, might help the agents solve the task, since the frozen encoders would provide more stable visual features for learning the embodied object recognition task (as demonstrated by Parisi et al., 2022). However, like all the other agents we tested, the agents failed to solve the task, largely learning background-dependent object representations.
>
> Third, we emphasize that we used both PPO and LSTM PPO policy networks in the paper.
>
> In future work, we will focus on inserting more powerful policy networks into the artificial agents. As the Reviewer suggests, it is possible that PPO and LSTM PPO policy networks are insufficient to solve this task. We hope researchers use our benchmark to find policy networks that can match the rapid and robust learning of chicks.
>
> **The Reviewer was concerned about the heuristic (imprinting) reward.**
>
> We agree with this concern. Our aim is to offer a challenging benchmark that requires deep RL algorithms to be trained and tested in the same environment as real animals in controlled-rearing experiments. To succeed at this task, researchers may need to use ML systems that use different rewards, like curiosity. In our revision, we emphasize that we designed the benchmark to allow researchers to use their own custom-designed intrinsic rewards to drive the agent’s behavior (i.e., researchers do not need to use our imprinting reward). This makes the NETT more flexible: researchers can plug different encoders, policy networks, and/or intrinsic rewards into the artificial agents. We hope this NETT will be useful for discovering reward structures that produce robust object segmentation in embodied ML systems.
>
> That said, we informally tried a number of different rewards and found that this imprinting reward produced behavior that best matched the imprinting behavior of newborn chicks. However, we agree with the reviewer that newborn animals learn through intrinsic motivations like curiosity, so building ML systems that learn like brains will ultimately require “turning off” our imprinting reward and using a more biologically plausible intrinsic reward.
>
> **The Reviewer argued that parts of the experimental description were unclear.**
>
> To fix this problem, we (1) improved the resolution of the figures, (2) clarified what we mean by low-signal-to-noise data including citation, and (3) clarified that Wood & Wood (2021a) collected the chick data used to create the NETT benchmark.

---

### Author Response · Authors · 2023-11-22
**Global Response**

Our paper introduces a matched-experience Newborn Embodied Turing Test (NETT) for comparing the object segmentation abilities of newborn chicks and machine learning (ML) algorithms. When raised (trained) in the same visual environment, we found that newborn chicks learn background-invariant object representations, whereas ML algorithms learn background-dependent representations. This NETT exposes core limitations in current ML algorithms in achieving brain-like object perception. We thus argue that this will be a useful benchmark to help researchers close the learning gap between brains and machines.

The Reviewers largely agreed that this is an interesting topic and that our NETT has value. The Reviewers also offered a number of helpful suggestions for improving the paper:

First, in the original submission, the artificial agents were trained for 1,000 episodes, raising the possibility that the agents could learn background-invariant object representations with more training. To test this possibility, we trained the artificial agents with 10X more episodes (10,000 episodes). As shown in Fig. 3G, the agents still learned background-dependent object representations, akin to agents trained for 1,000 episodes (Fig. 3B-F). Our conclusions thus generalize across a range of training times.

Second, in the original submission, it was not clear why the artificial agents failed. Did they fail because they had insufficient visual encoders or policy networks? To test whether the failure was due to the encoders, we inserted a variety of powerful pre-trained visual encoders into the agents, including DINO_V2 (Oquab et al., 2023), Ego4D (Radosavoic et al., 2022), and Segment Anything (Kirillov et al., 2023). These pre-trained models perform well on standard computer vision benchmarks for object segmentation; nevertheless, artificial agents equipped with these pretrained encoders still failed to solve the one-shot learning task solved by newborn chicks. As shown in Fig. S3, agents with pre-trained encoders also showed a background-dependent pattern of performance.

Third, we explored whether the policy networks could solve the task when learning from stable visual features, by pre-training the encoders using simulated first-person images from inside the virtual chamber (Fig. S2). We collected 80,000 first-person images for each of the six rearing conditions and used those images to train CNN visual encoders. We used a self-supervised contrastive learning objective that leverages the temporal structure of natural visual experience, without relying on arbitrary image augmentations or labels (Schneider et al., 2021). The encoders were pre-trained in the same environment in which we trained the artificial agents. We reasoned that training the encoder and policy networks sequentially, rather than simultaneously, might help the agents solve the task, since the frozen encoders would provide stable visual features for learning the embodied object recognition task. However, like all the other agents we tested, the agents failed to solve the task, learning background-dependent representations.

Fourth, in the original submission, the artificial agents only had one ‘eye’ (camera) with a limited field of view (FOV), whereas newborn chicks have two eyes. It is possible that this difference led to the divergent learning outcomes in the biological versus artificial agents. To allow researchers to explore this possibility, we modified the NETT so that researchers can select either a one-eyed or two-eyed agent (Fig. S4). The two-eyed agent was carefully constructed so that the FOV of each eye matches the FOV of newborn chicks.

Fifth, we note that while we used an imprinting reward to encourage the artificial agents to imprint, our benchmark is designed to allow researchers to use their own custom-designed intrinsic rewards to drive the agents' behavior (i.e., researchers do not need to use our imprinting reward). This makes the NETT more flexible: researchers can plug any intrinsic reward into the artificial agents and test whether that agent learns as efficiently as chicks.

We emphasize that researchers can flexibly control how artificial agents are trained in the NETT. Our NETT is designed to run with Stable Baseline 3, so researchers can vary (among other parameters) the number of training episodes, length of the training episodes, batch size, number of test episodes, length of test episodes, learning rate, visual encoder, policy network, and the image resolution of the eye(s). Finally, to help visualize and study machine and animal behavior through a common lens, the NETT allows users to record images from an overhead camera and from a camera capturing the agent's first-person views.

---

### Meta-Review · Area_Chair_W32h · 2023-12-05

**Metareview:**

This paper describes a grounded benchmark to compare artificial and biological systems. This benchmark aims to help researchers better understand the difference in capabilities between artificial and biological agents to inspire the development of AI systems with greater sample efficiency.

Except for one reviewer, there was general agreement that the proposed benchmark was valuable because it offers a controlled environment for the data diet fed to both systems during training. The benchmark appears to be timely, and the data collected from newborn chicks were deemed compelling and interesting.

One of the reviewers felt strongly that SOTA training approaches should have been incorporated into the evaluation, but this reviewer did not engage with the authors or other reviewers. The AC believes that the points raised by this reviewer were well addressed in the rebuttal, and hence, this negative review can be discarded. Two positive reviewers took part in the discussion to re-iterate their support for the paper to be accepted.

Overall the AC recommends the paper be accepted.

**Justification For Why Not Higher Score:**

The proposed benchmark will only appeal to a limited community of researchers interested in developing AI systems grounded in neuroscience.

**Justification For Why Not Lower Score:**

This is a novel and interesting benchmark which will appeal to some members of the ICLR community.

---

### Decision · Program_Chairs · 2024-01-16

Accept (poster)